# Finding Densest Subgraphs with Edge-Color Constraints

## ABSTRACT

Finding densest subgraphs is a classical graph-mining problem that has many applications in Web-data analysis, such as identifying groups of related Web documents, finding communities of users, detecting fraudulent behavior, and more. In this paper, we consider a variant of the densest subgraph problem in networks with single or multiple edge attributes. For example, in a social network, the edge attributes may describe the type of relationship between users, such as friends, family, or acquaintances, or different types of communication between users. For conceptual simplicity, we view the attributes as *edge colors*. The new problem we address is to find a *diverse densest subgraph* that fulfills given requirements on the numbers of edges of specific colors. When searching for a dense social network community, our problem will enforce the requirement that the community is diverse according to criteria specified by the edge attributes. We show that the decision versions for finding *exactly*, *at most*, and *at least* h colored edges densest subgraph, where h is a vector of color requirements, are NP-complete, for already two colors. For the problem of finding a densest subgraph with *at least* h colored edges, we provide a linear-time constant-factor approximation algorithm when the input graph is sparse. On the way, we introduce the related at least $h$ (non-colored) edges densest subgraph problem, show its hardness, and also provide a linear-time constant-factor approximation. In our experiments, we demonstrate the efficacy and efficiency of our new algorithms.

## CCS CONCEPTS

• **Information systems** → **Social networks**; • **Theory of computation** → *Graph algorithms analysis*.

## KEYWORDS

Density, Densest subgraph, Diversity, Social networks

**ACM Reference Format:**
Anonymous Author(s). 2018. Finding Densest Subgraphs with Edge-Color Constraints. In *Proceedings of Make sure to enter the correct conference title from your rights confirmation emai (Conference acronym 'XX)*. ACM, New York, NY, USA, 12 pages. https://doi.org/XXXXXXX.XXXXXXX

## 1 INTRODUCTION

Graph analysis plays a pivotal role in understanding the intricate structure of the World Wide Web, offering insights into the relationships and connections that underpin its vast digital landscape. Finding densest subgraphs is a classical graph-theoretic problem

**Figure 1: Example for the at least h colored edges densest subgraph problem in a toy social network with two relationship types. The subgraph induced by $S_1$ is the densest unconstrained subgraph. If we require the densest subgraph to contain at least four edges of type two (red dashed), the graph induced by $S_2$ is optimal.**

and one of the most fundamental issues in graph data mining and social network analyses [20, 29]. In one of its most basic versions of the *densest-subgraph problem* (DSP), we are given an undirected finite graph $G = (V, E)$, and the goal is to find a subset of nodes $S \subseteq V$ such that the induced subgraph maximizes the ratio between edges and nodes $|E(S)|/|S|$. Examples of the many Web-related applications are, e.g., community detection in social networks [11, 15, 40], real-time story identification [3], identifying malicious behavior in financial transaction networks [28] or link-spam manipulating search engines [19], and team formation in social networks [17, 35]. The problem also has applications in other domains such as, e.g., analyzing biological networks [36, 42], or general applications in data structures like indexing of reachability and distance queries [12]. Recently, the increasing interest in algorithms that ensure fairness or diversity [27, 33] has been extended to finding *diverse dense subgraphs*. Anagnostopoulos et al. [1] and Miyauchi et al. [34] discuss variants of the DSP that include fairness and diversity properties in graphs with respect to the node attributes.

**Our work:** We introduce new problem definitions for finding *edge-diverse* dense subgraphs in graphs with *categorical edge attributes*, which we, for conceptual simplicity, denote as edge colors. More specifically, we introduce the problems of finding a densest subgraph with *at least* **h** *colored* edges, where the vector **h** contains for each attribute, i.e., color, the minimum number of edges that are required to be in the solution. Similarly, we define two variants for *exactly* and *at most* **h** colored edges. Figure 1 shows a small toy example of a network containing two different relationship types. Computing the standard densest subgraph leads to the monochrome subgraph induced by $S_1$. To obtain a diverse subgraph that also contains edges of relationship type two, we apply the at least **h** colored edges densest subgraph variant and can identify the densest subgraph induced by $S_2$ that contains edges of relationship type two. We can apply our new problem in the following scenarios in Web-related networks.

*Web Graph Analysis:* Consider a large web graph in which nodes represent online articles published on websites or blogs and edges represent relations between articles. The edges can represent, e.g.,

citations, extensions, or hyperlinks between the websites. Additionally, the edges are annotated with meta-information further describing the relationship, e.g., shared topic, type of relationship between the articles, agreement, refusal, or sentiment between articles. Now, a typical task is to obtain a summary highlighting the most interconnected parts of the network [28, 29]. However, without further restrictions on the edge attributes, the resulting subgraph may completely ignore or exclude specific attributes that are not part of the unconstrained densest subgraph. Using the at least **h** colored densest subgraph, a user has the possibility to include specific attributes into the network summary by setting the corresponding entries in the requirement vector **h** to the minimum number of included relations between the articles.

*Online Social Network Analysis:* The participants of large-scale social networks are commonly connected to hundreds or even thousands of other users. Typically, user relationships are heterogenic and can be distinguished in, e.g., friendship, family membership, acquaintance, or work colleague [23]. Additionally, the strength of the relationship is often classified into weak and strong ties, where weak ties often have the capacity to bridge diverse social groups and facilitate the flow of information [22, 38]. By requiring specific numbers of edge attributes, we identify densest subgraphs related across multiple attribute dimensions. In addition to mere interconnections, the resulting dense subgraphs embody communities with diverse relationships. Moreover, identifying dense subgraphs with minimal specific relationships can be advantageous for subsequent tasks, such as content recommendation. For instance, a dense subgraph including many weak professional connections can form the foundation for recommending new professional contacts bridging into new social groups.

**Contributions:** Our contributions are the following.

(1) We introduce new variants of diverse densest subgraph problems in edge-colored graphs. We are interested in finding densest subgraphs that contain exactly $h_i$, at most $h_i$, or at least $h_i$ edges of color $i \in [\pi]$ where $\pi$ is the number of colors in the graph. We discuss variants of the problems in which each edge either has a single or multiple colors. We show that the corresponding decision problems are **NP**-complete.

(2) For the problem of at least **h** colored edges in sparse graphs we introduce a linear-time $O(1)$ approximation algorithm.

(3) As an additional result, we introduce the densest subgraph problem with at least $h$ (non-colored) edges, show that the problem is **NP**-hard as well, and also provide a linear-time $O(1)$ approximation algorithm.

(4) We evaluate our algorithms on real-world networks and demonstrate that (i) our approximation algorithms have very low relative approximation errors, in most cases under one percent, and (ii) are highly efficient computable.

Please refer to Appendix A for the omitted proofs.

## 2 RELATED WORK

**Finding densest subgraphs.** Finding densest subgraphs is a fundamental problem in network analysis and has a variety of applications. The problem has gained increasing interest in recent years, both in theoretical computer science and data-mining communities. An extensive review of the densest subgraph problem, its variants, properties, and algorithms is beyond the scope of this paper, so here we discuss only the most relevant work. For a recent survey on the topic, we refer the reader to Lanciano et al. [28].

The unconstrained version of the problem, when the density of a subgraph induced by a subset of vertices $S \subseteq V$ of a graph $G = (V, E)$ is defined as $d(S) = |E(S)|/|S|$, is solvable in polynomial time via max-flow computations [21]. For a more efficient but approximate solution, a linear-time greedy algorithm, which removes iteratively the node of the smallest degree and returns the best solution encountered, provides an approximation ratio equal to two [4, 9]. That type of greedy algorithm is often referred to as *peeling*. Recently, Chekuri et al. [10] provided an almost linear-time flow-based algorithm, approximating the densest subgraph problem within $(1 + \epsilon)$. Chekuri et al. [10] also analyzed an iterative peeling algorithm proposed by Boob et al. [7] and showed that it converges to optimality. Research has also focused on the problems of finding densest subgraphs with at most $k$ nodes (Dam$k$S), at least $k$ nodes (Dal$k$S), and exactly $k$ nodes (D$k$S). The D$k$S problem is **NP**-hard, even when restricted to graphs of maximum degree equal to 3 [16], and the best-known approximation ratio is $O(n^{1/4})$ [6]. With respect to the upper-bound variant, Khuller and Saha [26] showed that an $\alpha$-approximation for the Dam$k$S problem leads to an $\alpha/4$-approximation for D$k$S.

More related to our work is the Dal$k$S problem, which is also **NP**-hard [26]. Andersen and Chellapilla [2] designed a linear-time 1/3-approximation algorithm based on greedy peeling, while Khuller and Saha [26] provided two algorithms, both yielding a 1/2-approximation, using flow computations and solving an LP, respectively.

Finally, our work is related to finding densest subgraphs in multi-layer networks. Galimberti et al. [18] discussed the $k$-core decomposition and densest subgraph problems for multilayer networks and provided an approximation algorithm for a different formulation than the one we study in this paper. We experimentally compare our algorithm with the method of Galimberti et al. [18] and show that our approach finds denser subgraphs.

**Diverse densest subgraphs.** Two recent works consider diversity in finding densest subgraphs. Anagnostopoulos et al. [1] introduce the *fair densest subgraph problem*. The authors consider graphs with nodes labeled by two colors, and the goal is to find a subset of nodes that contains an equal number of colors. They show that their problem is at least as hard as the Dam$k$S problem. Moreover, they propose a spectral algorithm based on ideas by Kannan and Vinay [25]. In the second work, Miyauchi et al. [34] discuss generalizations of the problem introduced by Anagnostopoulos et al. [1]. They introduce two problem variants. The first problem guarantees that no color represents more than some fraction of the nodes in the output subgraph. The second problem is the "node version" of the problem we discuss, i.e., they study the densest subgraph problem, in which, given a vector of cardinality demands for each color class, the task is to find a densest subgraph fulfilling the demands.

Both of these works focus on node-attributed graphs, while we study the densest subgraph problem for edge-attributed graphs.

## 3 PROBLEM DEFINITIONS

We use $\mathbb{N}$ to denote the natural numbers (without zero). Furthermore, for $x \in \mathbb{N}$, we use $[x]$ to denote the set $\{1, \dots, x\}$. Vectors are

denoted in boldface, e.g., $\mathbf{h} \in \mathbb{N}^x$, and $h_i$ represents the $i$-th entry of $\mathbf{h}$. An undirected, simple graph $G = (V, E)$ consists of a finite set of vertices $V$ and a finite set $E \subseteq \{\{u, v\} \subseteq V \mid u \neq v\}$ of undirected edges. We define $n = |V|$ and $m = |E|$. An edge-colored graph $G = (V, E, c)$ is a graph with an additional function $c : E \rightarrow 2^{\mathbb{N}}$ assigning sets of colors to the edges. For notational convenience, we write $c(e) = i$ instead of $c(e) = \{i\}$ if edge $e \in E$ is assigned a single color. For $S \subseteq V$, we define $E(S) = \{\{u, v\} \in E \mid u, v \in S\}$ and $G(S) = (S, E(S))$ the subgraph induced by $S$.

**Definition 1.** *Given an edge-colored graph $G = (V, E, c)$, a number of colors $\pi \in \mathbb{N}$, and a vector $\mathbf{h} \in \mathbb{N}^\pi$, find a subset $S \subseteq V$ such that*

- *$E(S)$ contains at least $h_i$ edges with $i \in c(e)$ for all $i \in [\pi]$, and*
- *the density $d(S) = \frac{|E(S)|}{|S|}$ is maximized.*

Similarly, we define the *exactly* $\mathbf{h}$ and the *at most* $\mathbf{h}$ colored edges densest subgraph problem variants.

We can check the feasibility for the at least $\mathbf{h}$ and the exactly $\mathbf{h}$ colored edges variants in linear time by counting the occurrences of colors $i \in [\pi]$ at all edges in $G$. In the case of the at most $\mathbf{h}$ colored edges variant, the empty subgraph is a feasible solution. Hence, in the following, we do not make the feasibility check explicit and consider instances to be feasible.

**Complexity.** In contrast to the standard variant of the densest subgraph problem, which can be solved optimally in polynomial time, adding constraints on the numbers of colored edges makes the problems hard. Indeed, we show hardness already for the case that each edge is colored by one of only two colors.

THEOREM 1. *The decision versions of the exactly, at most, and at least $\mathbf{h}$ colored edges version are NP-complete.*

The proofs are based on reductions from the $k$-clique problem and are provided in detail in Appendix A.1.

## 4 APPROXIMATION IN SPARSE GRAPHS

In this section, we present a $O(1)$-approximation for at least $\mathbf{h}$ colored edges densest subgraph problem (in the following also denoted as ALHCEDGESDSP) for *everywhere sparse graphs*.

We call a graph $G = (V, E)$ sparse if $|E| = O(|V|)$. A graph $G = (V, E)$ is *everywhere sparse* if for any subset $V' \subseteq V$ the by $V'$ induced subgraph $G(V')$ is sparse. We first focus on the case that each edge in the graph is assigned a single color and discuss the general case in Section 4.3.

Our approximation for ALHCEDGESDSP is based on finding densest subgraphs with at least $h$ edges (ignoring colors of the edges). We define the problem as follows.

**Definition 2** (At least $h$ edges densest subgraph problem)**.** *Given a graph $G = (V, E)$ and $h \in \mathbb{N}$, find a subset $S \subseteq V$ such that $|E(S)| \geq h$, and the density $d(S) = \frac{|E(S)|}{|S|}$ is maximized.*

We denote the problem with ATLEASTHEDGESDSP and show that this problem without colors is already hard.

THEOREM 2. *The decision problem of the at least $h$ edges DSP is NP-complete.*

### 4.1 Solving the at Least $h$-Edges DSP

First, assume we have an algorithm for the at least $k$-nodes DSP problem. We can use it to solve the at least $h$-edges DSP problem (ATLEASTHEDGESDSP). To this end, let $\ell(h)$ a lower bound on the number of nodes of a graph with at least $h$ edges. For generality, define the lower bound $\ell(h, p)$ for graphs with up to $p$ parallel edges between two nodes and define $\ell(h) = \ell(h, 1)$ (we use the general version for the case of multigraphs as discussed in Section 4.3).

**Lemma 1.** *Let $G = (V, E)$ be a graph with $|E| \geq h$ and at most $p$ parallel edges between each pair of nodes. Then $\ell(h, p) = \frac{1}{2} + \frac{\sqrt{p^2 + 8hp}}{2p}$.*

PROOF. The number of nodes of a graph with $|E| = h$ is minimized if $G$ is complete and there are $p$ parallel edges between each pair of nodes, i.e., $h = p\binom{|V|}{2}$. Solving for $|V|$ leads to $\ell(h, p)$. □

---

**Algorithm 1:** Algorithm for ATLEASTHEDGESDSP

**Input:** Graph $G = (V, E)$ and $h \in \mathbb{N}$
**Output:** Densest subgraph with at least $h$ edges
1 **for** $i \in \{\ell(h), \ldots, n\}$ **do**
2     Compute the at least $i$ nodes DSP $S_i$
3 **return** $S_i$ with maximal density and at least $h$ edges

---

The following lemma establishes a connection between the at least $k$ nodes and at least $h$ edges DSP.

**Lemma 2.** *Given a graph $G$ and $h \in \mathbb{N}$. Let $k$ be the minimum number of nodes over all graphs that are densest subgraphs of $G$ with at least $h$ edges. Furthermore, let $S$ be an optimal solution for the at least $k$-nodes DSP problem in $G$. Then $S$ is also an optimal solution for the densest subgraph with at least $h$ edges.*

Based on Lemma 2 Algorithm 1 computes the solution of the at least $h$ edges DSP.

THEOREM 3. *Algorithm 1 is optimal for ATLEASTHEDGESDSP.*

PROOF. Algorithm 1 computes an optimal solution of the at least $i$-nodes DSP for each $i \in \{\ell(h), \ldots, n\}$. We know that the optimal solution of ATLEASTHEDGESDSP has at most $n$ nodes. Due to Lemma 2, Algorithm 1 discovers at least one optimal solution $S$ of the densest subgraph with at least $h$ edges. Finally, an optimal solution will be returned as Algorithm 1 returns the $S_i$ with maximal density and at least $h$ edges. □

Now, let $G$ be an everywhere sparse graph, and assume we have an $\alpha$-approximation algorithm for the at least $k$-nodes DSP problem. We can obtain an $O(1)$-approximation for the least $h$-edges DSP problem (ATLEASTHEDGESDSP).

THEOREM 4. *Algorithm 2 gives an $O(1)$-approximation for ATLEASTH-EDGESDSP.*

PROOF. Let $k$ be the minimum number of nodes over all graphs that are densest subgraphs with at least $h$ edges. And, let $d(S)$ be an $\alpha$-approximation of the at least $k$-nodes DSP. Then, from Lemma 2, it follows that $d(S)$ is also an $\alpha$-approximation of the at least $h$-edges DSP. However, $S$ may not be a feasible solution because it

---

**Algorithm 2:** Algorithm for atLeastHEdgesDSP

**Input:** Everywhere sparse graph $G = (V, E)$ and $h \in \mathbb{N}$
**Output:** Approx. of densest subgraph with at least $h$ edges

1 **for** $i \in \{\ell(h), \ldots, n\}$ **do**
2    $\alpha$-approximate the at least $i$-nodes DSP and obtain
     solution $S_i$
3    **if** $G(S_i)$ *does not have at least $h$ edges* **then**
4      add edges to $G(S_i)$ to obtain $G_i'$ with at least $h$ edges

5 **return** $G_i'$ with maximum density

---

may have less than $h$ edges, i.e., $|E(S)| < h$. Assume that for a constant $c_1 \in \mathbb{N}$ the result of the $\alpha$-approximation, it holds that $c_1|E(S)| \geq |E(S^*)| \geq h$, where $S^*$ is the optimal solution for the at least $i$-nodes DSP. Algorithm 2 adds the possibly missing edges to obtain the subgraphs $G_i' = (S_i', E_i')$ (line 4). At most $h$ edges and $2h$ nodes are added, and $|S_i'| \leq |S| + 2h \leq |S| + 2c_1|E(S)| \leq c|S|$ where the last inequality holds due to the everywhere sparseness property of $G$ for a large enough constant $c \in \mathbb{N}$. Consequently,

$$d(S') = \frac{|E(S')|}{|S'|} \geq \frac{|E(S)|}{c|S|} = \frac{1}{c}d(S) \geq \frac{1}{c\alpha}d^*,$$

where $d^*$ is the optimal density. $\square$

The assumption that $c_1|E(S)| \geq |E(S^*)| \geq h$ with $c_1 \in \mathbb{N}$ holds for example for the 2-approximation and 3-approximation algorithms provided by Khuller and Saha [26] and Andersen and Chellapilla [2], respectively.

The running time complexity of Algorithm 2 is in $O(n(T_{\text{appr}} + h))$ with $T_{\text{appr}}$ being the running time of the at least $i$ nodes DSP approximation as we have $n$ rounds and in each round we call the approximation and have to add at most $h$ edges to obtain $G_i'$. Using the 3-approximation algorithm by Andersen and Chellapilla [2] based on the $k$-core computation, we can obtain an approximation with total running time in $O(n + m)$.

---

**Algorithm 3:** Approximation for atLeastHEdgesDSP

**Input:** Everywhere sparse graph $G = (V, E)$ and $h \in \mathbb{N}$
**Output:** Approx. of densest subgraph with at least $h$ edges

1 $G_0 \leftarrow G$, $i \leftarrow 0$, and $i_{\max} \leftarrow 0$
2 **while** $|E(G_i)| \geq h$ **do**
3    $i_{\max} \leftarrow i$
4    Increment $i$
5    Let $v_i$ be a node with minimum degree
6    $G_i \leftarrow G_{i-1} \setminus \{v_i\}$      // remove $v_i$ and all incident edges

7 **return** $G_i$ for $i \in \{0, \ldots, i_{\max}\}$ with maximum density

---

THEOREM 5. *Algorithm 3 is a $O(1)$-approximation for* atLeastH-EdgesDSP *with running time in $O(n + m)$.*

PROOF. Algorithm 3 peels away low degree nodes and thus obtains $G_0, \ldots, G_{i_{\max}}$. Assume we similarly computed the remaining graphs $G_{i_{\max}+1}, \ldots, G_n$ (as in the standard $k$-core decomposition). Andersen and Chellapilla [2] showed that for each possible $j \in [n]$

one of the $G_0, \ldots, G_{n-j}$ is a 3-approximation for the at least $j$-nodes DSP, and the number of edges is at least $1/3$ of the optimal solution.

Now, let $k$ be the minimum number of nodes over all graphs that are densest subgraphs with at least $h$ edges. And, let $d(S)$ be the 3-approximation of the at least $k$-nodes DSP. Then, from Lemma 2, it follows that $d(S)$ is also an 3-approximation of the at least $h$-edges DSP. There are two cases:

(1) $G(S)$ contains at least $h$ edges, i.e., corresponds to one of the graphs in $\{G_0, \ldots, G_{i_{\max}}\}$. In this case, we are done, and $G(S)$ is a 3-approximation for atLeastHEdgesDSP.

(2) $G(S)$ contains less than $h$ edges (but at least $1/3$ of the optimal solution), i.e., corresponds to a graph $H$ in $G_{i_{\max}+1}, \ldots, G_{n-k}$. In this case, we need to add edges such that $G(S)$ is feasible. Let $S'$ be the resulting vertex set. With similar arguments as in the proof of Theorem 4, it follows that $d(S') \geq \frac{1}{c}d^*$ for a large enough constant $c \in \mathbb{N}$ and with $d^*$ being the optimal density. Now, as we can choose the edges that we add to achieve feasibility, we choose exactly the edges in $E(G_{i_{\max}}) \setminus E(H)$ such that $G(S') = G_{i_{\max}}$. Note that we add in the worst case at most $2h$ nodes. Hence, $G_{i_{\max}}$ is a $c$-approximation.

As Algorithm 3 returns the $G_i \in \{G_0, \ldots, G_{i_{\max}}\}$ with maximum density, in either case, we obtain a $O(1)$-approximation for atLeastHEdgesDSP. The running time complexity of Algorithm 3 is equal to the peeling-based $k$-core decomposition, which is $O(|V| + |E|)$ as shown by Batagelj and Zaversnik [5]. $\square$

## 4.2 Approximation of the at Least h Colored Edges DSP

We now can use Algorithm 2 or Algorithm 3 to obtain a $O(1)$-approximation for the alhcEdgesDSP problem in everywhere sparse graphs as shown in Algorithm 4.

---

**Algorithm 4:** Approximation for alhcEdgesDSP

**Input:** Everywhere sparse graph $G = (V, E)$ and $\mathbf{h} \in \mathbb{N}^\pi$
**Output:** Approx. of densest subgr. with at least $h_i$ edges of
     color $i \in [\pi]$

1 Approximate the densest subgraph $G' = (V', E')$ with at
   least $\sum_{i=1}^\pi h_i$ edges
2 Let $f_i$ be the number of edges of color $i$ in $G'$ for $i \in [\pi]$
3 $G'' \leftarrow G'$
4 **for** $1 \leq i \leq \pi$ **do**
5    Add $\max\{0, h_i - f_i\}$ edges of color $i$ to $G''$

6 **return** $G''$

---

THEOREM 6. *Algorithm 4 gives an $O(1)$-approximation for* alhc-EdgesDSP.

PROOF. Let $G'' = (V'', E'')$ be the final resulting subgraph. Because atLeastHEdgesDSP relaxes alhcEdgesDSP, we have $d(V') \geq \frac{d^*}{\alpha}$ using an $\alpha$-approximation for the at least $h$ edges subgraph, and where $d^*$ is the optimal density of atLeastHEdgesDSP. We add in total

$$\ell = \sum_{i=1}^\pi \max\{0, h_i - f_i\} \leq \sum_{i=1}^\pi h_i \leq |E'|$$

edges to $G'' = (V'', E'')$ to ensure feasibility for ALHCEDGESDSP. Because each edge adds at most two vertices and $|V''| \leq |V'| + 2\ell \leq |V'| + 2|E'| \leq c|V'|$ with $3 \leq c \in \mathbb{N}$ and it follows

$$d(V'') = \frac{|E''|}{|V''|} \geq \frac{|E'|}{c|V'|} = \frac{1}{c}d(V') \geq \frac{1}{c\alpha}d^*. \qquad \square$$

In Algorithm 4, after obtaining $G'$ from Algorithm 3 we might need to add missing edges. Note that $G'$ is node-induced, and we have to insert at least one new node. Of course, adding all edges of the new node to $G'' = (V'', E'')$ only improves the density. To add the missing nodes, we store the nodes that are removed in the call of Algorithm 3 that lead to missing edges. Therefore, we use a slightly modified version Algorithm 3 as a subroutine to approximate the at least $h$ edges DSP. During the iterations of the while-loop in Algorithm 3, vertices and their incident edges are removed from the graph. Here, if in iteration $i$ we have to remove an edge $e = \{u, v\}$ with $c(e) = c$ and the remaining edges of color $c$ is smaller than $h_c$, i.e., removing $e$ leads to a deficit of color $c$ edges, then we add both endpoints of $e$ to a set $B_i$, where $B_i = B_{i-1} \cup \{u, v\}$ (and $B_0 = \emptyset$). At the end of the subroutine, we return $G_i = (V_i, E_i)$ for $i \in \{0, \ldots, i_{\max}\}$ with maximum density and the corresponding set $B_i$. Note that $|B_i| \leq 2\ell$, i.e., for each missing colored edge, at most two nodes are in $B_i$. Therefore, we can just add the nodes in $B_i$ and return as a final result of Algorithm 4 the graph $G'' = (V' \cup B_i, E')$, which leads to a running time of $O(n + m)$ of Algorithm 4.

## 4.3 Graphs with Multiple Edge Colors

Up to this point, our focus was on graphs with single-colored edges. However, in practical contexts, edges often bear multiple distinct colors, each representing diverse aspects of node interactions. Discovering the densest subgraph that illuminates these varying types of node interactions holds intrinsic value. To accommodate this case, we transform a simple graph with multiple edge colors into a colored multigraph, where each edge is assigned a single color.

An undirected edge-colored multigraph $M$ is defined as a tuple $M = (V, E_M, f, c_M)$ where $V$ is a finite set of vertices, $E_M$ is a finite set of edges, $f : E_M \rightarrow \{\{u, v\} : u, v \in V\}$ is a function mapping edges to pairs of vertices, and $c_M : E_M \rightarrow \mathbb{N}$ is a function assigning to each edge a single color. If $e_1, e_2 \in E_M$ and $f(e_1) = f(e_2)$, then we call $e_1$ and $e_2$ multiple or parallel edges. For a subset $S$ of vertices $S \subseteq V$, we define $E_M(S) = \{e \in E_M \mid f(e) \subseteq S\}$ and $M(S) = (S, E_M(S), f, c_M)$ the multigraph induced by $S$. The density of $M(S)$ is defined as $d(S) = \frac{|E_M(S)|}{|S|}$.

Given an edge-colored simple graph $G = (V, E, c)$ with multiple edge colors, we construct an associated multigraph, denoted as $M = (V, E_M, f, c_M)$, sharing the same set of nodes, $V$. For each edge $e \in E$ and for each color $i \in c(e)$, we introduce an edge $e'$ into $E_M$ and assign $c_M(e')$ to be equal to $i$. If $G$ is everywhere sparse, then it follows that $M$ is also everywhere sparse if we consider the number of distinct colors $\pi$ to be a small constant, which is commonly the case for real-world networks (see, e.g., Table 1). Furthermore, note that the density of the obtained multigraph can be larger than that of the original graph. By counting the parallel edges separately, we account for the fact that edges with many colors have higher importance as they cover more of the color requirements compared to edges with single or few colors.

The insights and results derived in previous sections extend seamlessly to this multigraph framework, resulting in Algorithm 5 and the following theorem.

THEOREM 7. *Algorithm 5 gives an $O(1)$-approximation for ALHC-EDGESDSP on graphs with multiple edge colors.*

---

**Algorithm 5:** Approximation for ALHCEDGESDSP on graphs with multiple edge colors

---

**Input:** Everywhere sparse graph $G = (V, E, c)$ and $\mathbf{h} \in \mathbb{N}^\pi$
**Output:** Approximation of densest subgraph with at least $h_i$ edges of color $i$ for $i \in [\pi]$

1 Transform $G = (V, E, c)$ into multigraph $M = (V, E_M, f, c_M)$
2 Use Line 1-3 from Algorithm 4 applied to $M$
3 **while** *there exists $i \in [\pi]$ such that $f_i < h_i$* **do**
4      choose two nodes $u, v \in V$ that are connected by an edge of color $i$ and add all parallel edges between $u, v$ to $G''$
5      Update $f_i$ for $i \in [\pi]$
6 **return** $G''$

---

## 5 EXPERIMENTS

In this section, we evaluate our algorithms in terms of their efficacy and efficiency by discussing the following **research questions:**

**Q1:** How does our approximation for at least $h$ edges problem perform in terms of approximation quality?
**Q2:** How does increasing $h$ affect the density and running time?
**Q3:** How are the colors in the data sets and the unconstrained densest subgraphs distributed?
**Q4:** How does our approximation for the at least $\mathbf{h}$ color edges problem perform in terms of approximation quality?
**Q5:** How do increasing color requirements affect the densities?
**Q6:** How is the efficiency of our approximation for the at least $\mathbf{h}$ color edges problem in terms of running time?

Additionally, we discuss in Section 5.2, as a use case, the identification of densest subgraphs that contain publications at popular data mining conferences in a coauthor graph.

**Algorithms:** We use the following algorithms for our evaluation.

- ATLEASTHAPPROX is the implementation of Algorithm 4 for the at least $h$ edges DSP.
- COLAPPROX is the implementation of Algorithm 4 for the at least $h$ colored edges DSP.
- ATLEASTHILP and COLILP are the exact integer linear programs for finding the densest subgraph with at least $h$ edges and the densest subgraph with at least $\mathbf{h}$ colored edges, respectively. The ILPs are provided in Appendix B.
- HEURISTIC is a new baseline algorithm. As there is no baseline available for the at least $\mathbf{h}$ colored edges problem (Definition 1), we introduce a heuristic, serving for comparison and benchmarking. Given an edge colored graph $G = (V, E, c)$, the heuristic peels away nodes with the lowest degree as long as all color requirements are fulfilled.
- MLDS is a state-of-the-art approximation for the multilayer densest subgraph problem defined by Galimberti et al. [18]. The authors provided the source code.

We implemented our algorithms in C++ using GNU g++ compiler 11.4.0. with the flag −O3. The ILPs were implemented in Python 3.9 and ran with Gurobi 9.1.2. The experiments ran on a single machine with an Intel i5-1345U CPU and 32GB of main memory. The source code is available at https://gitlab.com/webconf24/ecdsp.

**Data Sets:** We use twelve real-world data sets from different domains and a wide range of different numbers of attributes, i.e., colors. Table 1 gives an overview of the statistics. We provide detailed descriptions in Appendix C.

**Table 1: Statistics of the data sets. $d(S^*)$ denotes the density of the unconstrained optimal densest subgraph and $\psi(G)$ denotes the maximal number of colors per edge.**

| Data set | $|V(G)|$ | $|E(G)|$ | $d(S^*)$ | #Colors | $\psi(G)$ | Category | Ref. |
|---|---|---|---|---|---|---|---|
| AUCS | 61 | 353 | 6.2 | 5 | 5 | Multilayer social | [32] |
| Hospital | 75 | 1 139 | 16.3 | 5 | 5 | Temporal face-to-face | [41] |
| HtmlConf | 113 | 2 196 | 20.5 | 3 | 3 | Temporal face-to-face | [24] |
| Airports | 417 | 2953 | 16.5 | 37 | 5 | Multilayer transportation | [8] |
| Rattus | 2 634 | 3 677 | 3.7 | 6 | 4 | Multilayer biological | [13] |
| FfTwYt | 6 401 | 60 583 | 39.5 | 3 | 3 | Multilayer social | [14] |
| Knowledge | 14 505 | 210 946 | 34.8 | 30 | 4 | Knowledge graph | [39] |
| HomoSap | 18 190 | 137 659 | 38.5 | 7 | 5 | Multilayer biological | [18] |
| Epinions | 131 580 | 592 013 | 85.6 | 2 | 2 | Signed (trust/no trust) | [30] |
| DBLP | 344 814 | 1 528 399 | 57.0 | 168 | 21 | Multilayer collaboration | [31] |
| Twitter | 346 573 | 1 088 260 | 45.1 | 2 | 2 | Signed (fact/non-fact) | [37] |
| FriendFeed | 505 104 | 18 319 862 | 500.1 | 3 | 3 | Multilayer social | [14] |

## 5.1 Results and Discussion

**Q1–Approximation quality of AtLeastHApprox.** To evaluate the approximation error of AtLeastHApprox, we computed the at least $h$ edges DSP for the AUCS, Hospital, and HtmlConf data sets using AtLeastHApprox and the exact ILP approach (AtLeastHILP) for all values of $w < h \leq |E(G)|$ with $w$ is the number of edges in the optimal unconstrained DSP. Table 2 shows the percentage of runs that were optimal, i.e., relative approximation error of zero, the percentage of the runs with a relative approximation error of at most one, and the statistics of the relative approximation errors in percent (%) for the runs that were not optimal. For Hospital and HtmlConf over 90% and 80%, respectively, of the instances are solved perfectly by AtLeastHApprox, and all instances are solved with a relative error of less than one. In the case of the AUCS data set, this value is lower with 38.9%. However, here, the mean and median relative approximation errors are also less than one percent, and more than 93% of the instances are solved with an error of at most one. The maximum relative error is at 1.24%.

**Table 2: Results and relative approximation errors in percent (%) for the at least $h$ edges DSP.**

| | | | Relative approximation errors (%) | | | |
|---|---|---|---|---|---|---|
| Data set | Opt. solved (%) | Within 1% err. (%) | Mean | Std. dev. | Median | Max. |
| AUCS | 38.9 | 93.1 | 0.59 | 0.31 | 0.42 | 1.24 |
| Hospital | 91.7 | 100 | 0.44 | 0.23 | 0.47 | 0.81 |
| HtmlConf | 83.0 | 100 | 0.14 | 0.13 | 0.10 | 0.83 |

**Q2–Density and running times for increasing $h$.** We computed the at least $h$-edges DSP where we choose $h = w + i$ with $w$ being the number of edges in the unconstrained DSP and $1 \leq i \leq |E| - w$.

Figure 2 shows the results for AUCS and FriendFeed, and Figure 7 in the appendix shows results for the remaining data sets. In all data sets, the densities of the subgraphs $S_i$ strongly decrease for larger $i$. As the size of $h$ increases, the peeling process can stop earlier, which leads to shorter running times: Table 3 shows the running times for the four largest data sets and increasing ratios $r$ such that $i = r \left( |E| - w \right)$. The reported running times are in seconds, and the mean values and standard deviations are over ten repetitions. As expected, the running time decreases with increasing value of $i$.

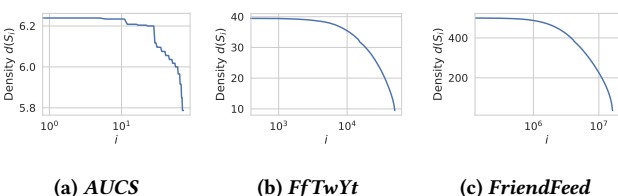

(a) AUCS      (b) FfTwYt      (c) FriendFeed

**Figure 2: The density computed with AtLeastHApprox for increasing numbers of required edges.**

**Table 3: Mean running times and standard deviations of AtLeastHApprox in seconds (s).**

| | $i = r \cdot (|E| - w)$ | | | | | | | | |
|---|---|---|---|---|---|---|---|---|---|
| Data set | $r = 0.1$ | $r = 0.2$ | $r = 0.3$ | $r = 0.4$ | $r = 0.5$ | $r = 0.6$ | $r = 0.7$ | $r = 0.8$ | $r = 0.9$ |
| Epinions | 0.31±0.0 | 0.28±0.0 | 0.25±0.0 | 0.22±0.0 | 0.19±0.0 | 0.16±0.0 | 0.14±0.0 | 0.10±0.0 | 0.07±0.0 |
| DBLP | 1.28±0.0 | 1.17±0.0 | 1.04±0.0 | 0.92±0.0 | 0.81±0.0 | 0.69±0.0 | 0.56±0.0 | 0.43±0.0 | 0.29±0.0 |
| Twitter | 0.84±0.0 | 0.76±0.0 | 0.69±0.0 | 0.63±0.0 | 0.56±0.0 | 0.49±0.0 | 0.42±0.0 | 0.32±0.0 | 0.19±0.0 |
| FriendFeed | 18.99±0.5 | 16.73±0.4 | 14.45±0.4 | 12.31±0.5 | 9.85±0.3 | 7.74±0.3 | 5.66±0.2 | 3.90±0.4 | 2.02±0.1 |

**Q3–Distribution of Colors in Unconstrained DSP.** First, we empirically verify the necessity of diversity in edge-colored graphs and the densest subgraphs by assessing the distribution of colors in the graphs and densest subgraphs. The findings consistently show that the fractions of the different colors are not equal in the data sets. Furthermore, often, the distribution of the colors in the unconstrained DSP differs significantly from the distribution in the complete graph. Figure 3 shows the distributions of the Knowledge and all data sets with two colors. For example, in Figure 3a and Figure 3b show the color distributions in the Knowledge data set and the (unconstrained) densest subgraph. The distribution between the graph and the DSP differs for most colors. Similarly, we see in the bichromatic data sets significant differences in the fractions of colors between the complete graph and the densest subgraph. Hence, even if there are many edges of a specific color in a graph, this does not generally mean that the densest subgraph contains a particular number of edges of this color. This validates the motivation for our new problems and algorithms. Furthermore, even if the distributions of the colors in the graph and the DSP are similar, we might want to specifically find a subgraph with specific numbers of edges of given colors.

**Q4–Approximation quality of ColApprox.** To evaluate the approximation quality of ColApprox, we first computed the at least $h$ colored edges DSP for the AUCS, Hospital, and HtmlConf data sets using Heuristic, ColApprox, and the exact ILP approach (ColILP). We chose 100 random problem instances. To this end, let

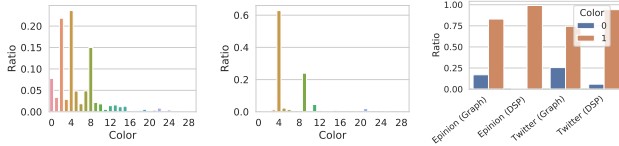

**(a)** *Knowledge* **(Graph)  (b)** *Knowledge* **(DSP)     (c) Binary data sets**

**Figure 3: The distributions of colors in various data sets.**

$f_i$ and $g_i$ be the number of edges of color $i$ in the unconstrained DSP and graph $G$, respectively. For each color $c \in [\pi]$, we chose $h_c$ uniformly at random from the interval $[f_c, g_c]$. Furthermore, let $\lambda = \frac{\sum_{c \in [\pi]} h_c}{\sum_{c \in [\pi]} (g_c - f_c)}$ denote the fraction of edges that are required from all possible additional edges. Figure 4 shows the densities of the solved problem instances with respect to $\lambda$. Moreover, Table 4 shows the statistics of the relative approximation errors in percent as well as the percentage of instances solved optimally or within a relative error of at most one. We see that ColApprox solves both more instances optimally and in the range of an error of at most one than Heuristic. The relative approximation errors are generally lower for ColApprox with mean values (much) smaller than one and maximum values of at most 1.89%.

**Table 4: Results and relative approximation errors in percent (%) for the at least h colored edges DSP.**

| Algorithm | Data set | Opt. solved (%) | Within 1% err. (%) | Relative approximation errors (%) | | | |
|---|---|---|---|---|---|---|---|
| | | | | Mean | Std. dev. | Median | Max. |
| Heuristic | *AUCS* | 21 | 76 | 0.65 | 0.82 | 0.27 | 3.58 |
| | *Hospital* | 1 | 19 | 2.64 | 1.94 | 2.14 | 9.10 |
| | *HtmlConf* | 6 | 62 | 0.99 | 0.87 | 0.79 | 3.69 |
| ColApprox | *AUCS* | 22 | 95 | 0.30 | 0.34 | 0.15 | 1.30 |
| | *Hospital* | 7 | 74 | 0.76 | 0.41 | 0.72 | 1.74 |
| | *HtmlConf* | 9 | 86 | 0.46 | 0.40 | 0.36 | 1.89 |

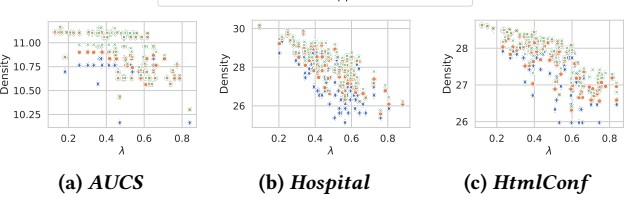

**(a)** *AUCS*                **(b)** *Hospital*                **(c)** *HtmlConf*

**Figure 4: Comparison of the heuristic, approximation algorithm, and exact ILP.**

**Q5–Density for increasing color requirements.** We computed the densities of the densest subgraph for increasing color requirements using ColApprox and Heuristic. To this end, we first computed the unconstrained DSP $H = (S, F)$. Let $f_c$ be the number edges of colors $c \in [\pi]$ in $F$, and $t_c$ be the total number of color $c \in [\pi]$ edges in $G$. For each color $c \in [\pi]$, we split define $r_c = (t_c - f_c)/10$. We then defined $h_c^i = i \cdot r_c$ for $i \in [10]$, leading to ten color requirement vectors $h^i$ with $i \in [10]$. Figure 5 shows the densities computed with ColApprox and Heuristic. For increasing numbers of required edges, the densities decrease. Compared to the Heuristic, our approximation algorithm ColApprox leads to higher or

at least as high densities for all data sets. For some data sets, e.g., *AUCS*, *HtmlConf*, *Epinions*, and *FriendFeed*, the Heuristic performs similarly good as our approximation. The reason is that for these data sets, all required colored edges are in the densest subgraphs. As the heuristic peels away nodes while the color requirements are not violated, the densest, or close to the densest, subgraph can be obtained in many cases. Also see **Q4** for a comparison between Heuristic and ColApprox with the optimal solutions, showing that for random instances ColApprox consistently outperforms Heuristic. Furthermore, Heuristic is bound to fail if the color requirements are violated early in the peeling process. In the following, we additionally show how the baseline fails in this case. To this end, we modified each data set by adding two nodes connected by a single edge of a new additional color. The results for Heuristic are shown in Figure 5 labeled Heuristic*. Because the nodes of the additional edge have a degree of one, the heuristic will try to remove them early on. But because the color of the new edge is required in the solution, the nodes cannot be removed, and Heuristic stops processing the graph, leading to much lower densities compared to ColApprox. For ColApprox, the density only changes minimally by the one additional edge and two additional nodes that need to be considered.

**Q6–Running times.** Table 5 shows the mean running times and standard deviations for ten repetitions of computations of the densities for $i \in \{2, 4, 6, 8\}$ where $i$ and the color requirements are chosen as in **Q5**. We show the results in seconds for the four largest data sets. The running times are only a fraction of a second for all other data sets. For both Heuristic and ColApprox the running time decreases for increasing $i$ and larger requirements of the colors. In the case of Heuristic, the reason is that the higher the requirements, the earlier the algorithm encounters a vertex whose removal would lead to a color deficit, and it stops. For ColApprox, the reason is that the higher the total color requirements, the earlier the subroutine that finds the at least $h$ edges densest subgraph can stop the peeling process.

**Table 5: Running times in seconds (s) for computing the at least h colored edges DSP.**

| Data set | $i = 2$ | | $i = 4$ | | $i = 6$ | | $i = 8$ | |
|---|---|---|---|---|---|---|---|---|
| | Heuristic | ColApprox | Heuristic | ColApprox | Heuristic | ColApprox | Heuristic | ColApprox |
| *Epinions* | 0.45±0.0 | 0.45±0.0 | 0.37±0.0 | 0.39±0.0 | 0.33±0.0 | 0.33±0.0 | 0.28±0.0 | 0.26±0.0 |
| *DBLP* | 1.25±0.0 | 1.74±0.0 | 1.01±0.0 | 1.48±0.0 | 0.94±0.0 | 1.21±0.1 | 0.84±0.1 | 0.88±0.0 |
| *Twitter* | 1.00±0.0 | 1.21±0.0 | 0.86±0.0 | 1.07±0.0 | 0.74±0.0 | 0.91±0.0 | 0.61±0.0 | 0.69±0.0 |
| *FriendFeed* | 18.85±0.2 | 21.06±0.3 | 16.69±0.2 | 16.50±0.2 | 13.97±0.2 | 12.11±0.2 | 10.25±0.1 | 8.25±0.1 |

## 5.2 Use Case: Diverse Coauthorship

In the following, we use a subgraph of the *DBLP* data set in which edges describe the coauthorship of publications published at one of ten data mining conferences. The network contains in total 44 823 nodes and 170 659 edges, each colored with one of ten colors representing the conference, and we are interested in finding densest subgraphs that contain publications of all conferences. Another view is that edges of color $c$ belong to layer $c$ of a multilayer graph with ten layers. Hence, we want to obtain densest subgraphs of the multilayer coauthor graph that contain at least $h_i$ edges in layer $i$. We compare our approximation algorithm ColApprox to the algorithm for the multilayer densest subgraph problem Mlds. The

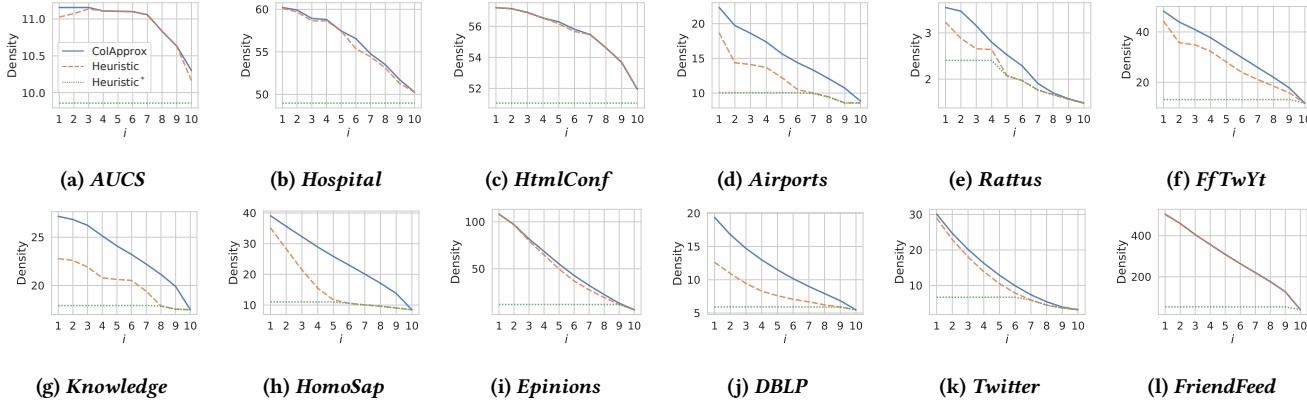

(a) *AUCS*    (b) *Hospital*    (c) *HtmlConf*    (d) *Airports*    (e) *Rattus*    (f) *FfTwYt*

(g) *Knowledge*    (h) *HomoSap*    (i) *Epinions*    (j) *DBLP*    (k) *Twitter*    (l) *FriendFeed*

**Figure 5: Densities for increasing color requirements (the common legend is shown in (a)).**

multilayer densest subgraph problem is defined as finding a subset $S \subseteq V$ such that

$$\max_{\hat{L} \subseteq L} \min_{\ell \in \hat{L}} \frac{|E_\ell(S)|}{|S|} \cdot |\hat{L}|^\beta,$$

is maximized, where $\beta \in \mathbb{R}$ is a parameter controlling the importance of adding few or many layers and $E_\ell$ are the edges in layer $\ell$ [18]. We computed the unconstrained densest subgraph, the multilayer densest subgraph for $\beta \in \{1, 2.2, 5\}$, where we chose the values of $\beta$ by increasing in 0.1 steps starting from one until all layers are in the densest subgraph. Only for the values of 2.2 and five do the results change. Furthermore, we use the at least **h** colored edges densest subgraph with the following color requirements. Let $t_c$ be the total number of color $c \in [\pi]$ edges in $G$. For each color $c \in [10]$, we define $h_c = t_c/\tau$ with $\tau \in \{10, 100, 1000\}$. Table 6 shows the results of the by $S$ induced subgraphs, including the density, where we use the standard definition of density, i.e., $d(S) = |E|/|S|$. For $\beta = 1$, the result of MLDS equals the unconstrained DSP. For $\beta = 2.2$, nine of the ten layers are included. Figure 6a shows the distribution of the publications. There are no publications from the *KDD* conference in the densest subgraph, and the density dropped significantly to 6.1 from the initial 20. Further increasing $\beta$ to five leads finally leads to a densest subgraph containing publications from all conferences; however, the density further dropped to 4.5. Moreover, the *KDD* conference is still underrepresented with only one publication (see Figure 6b). For our COLAPPROX, we obtain the densest subgraphs with 1/1000th, 1/100th, and 1/10th of the edges of each color while obtaining higher density values. Figure 6c and Figure 6d show the color distributions for $\tau = 1000$ and $\tau = 100$, respectively.

**Table 6: Computing densest coauthor subgraphs.**

| Algorithm | | Density | Nodes | Edges | Layers |
|---|---|---|---|---|---|
| Unconstrained DSP | | 20.0 | 41 | 820 | 2 |
| MLDS | $\beta = 1$ | 20.0 | 41 | 820 | 2 |
| | $\beta = 2.2$ | 6.1 | 232 | 1407 | 9 |
| | $\beta = 5$ | 4.5 | 396 | 1766 | 10 |
| COLAPPROX | $\tau = 1000$ | 8.7 | 137 | 1194 | 10 |
| | $\tau = 100$ | 9.5 | 409 | 3890 | 10 |
| | $\tau = 10$ | 9.4 | 2692 | 25324 | 10 |

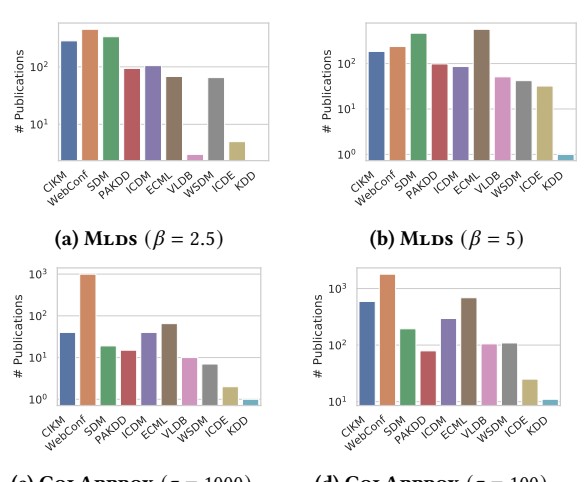

(a) MLDS ($\beta = 2.5$)    (b) MLDS ($\beta = 5$)

(c) COLAPPROX ($\tau = 1000$)    (d) COLAPPROX ($\tau = 100$)

**Figure 6: Distributions of conferences in the densest coauthor subgraphs computed with MLDS and COLAPPROX.**

## 6 CONCLUSION AND FUTURE WORK

We introduced new variants of diverse densest subgraph problems in networks with single or multiple edge attributes. We established the NP-completeness of decision versions for finding *exactly, at most*, and *at least* **h** colored edges densest subgraphs, even for just two colors. Furthermore, we presented a linear-time constant-factor approximation algorithm for the problem of finding a densest subgraph with at least **h** colored edges in sparse graphs. As an additional result, we introduced the related at least $h$ non-colored edges densest subgraph problem and provided a linear-time constant-factor approximation for it. Our experimental results validated the practical efficacy and efficiency of the proposed algorithms on a wide range of real-world graphs.

Future work directions include improving the approximation results for non-sparse graphs and introducing algorithms for the exact and at most **h** variants. Additionally, we plan to extend the problems and algorithms to dynamic settings where the topology and edge attributes of graphs can change over time, enabling us to better adapt to the evolving nature of the World Wide Web.

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

# A OMITTED PROOFS

In this section, we provide the omitted proofs.

## A.1 Proofs of Section 3

First, we provide the NP-completeness results for the decision versions of our edge-diverse densest subgraph problems.

THEOREM 8. *The decision version of the exactly* **h** *colored edges version (EHCEDGESDSPDEC) is* **NP**-*complete*.

PROOF. We show the result for the special case of two colors, i.e., $\pi = 2$. In this case, the decision version EHCEDGESDSP asks to decide if there is a subset $S$ such that the induced subgraph has exactly $h$ color $\sigma \in [2]$ edges and a density of at least $\alpha$. It is clearly in **NP**. We use a reduction from $k$-clique. Given an instance $(G, k)$ of $k$-clique, we construct the following instance of EHCEDGESDSPDEC:

- Let $G' = (V', E')$ with $V' = V \cup \{t\}$ and $E' = E \cup \{\{t, v\} \mid v \in V\}$, and $c(e) = 2$ if $e \in E' \cap E$, and $c(e) = 1$ otherwise,

- furthermore, let $\sigma = 1$, $h = k$, and $\alpha = \frac{\binom{k}{2} + k}{k+1} = \frac{k}{2}$.

Now, if $G$ contains a $k$-clique, then EHCEDGESDSPDEC has a *yes* answer. Because $t$ is connected to all $v \in V$ in $G'$, i.e., also to a set

$C$ containing $k$ nodes of a $k$-clique, we find a subgraph $S = C \cup \{t\}$ containing a $(k+1)$-clique. Because $G(S)$ is complete, $d(S) = \frac{\binom{k}{2}+k}{k+1}$.

For the other direction, assume eHcEdgesDSPdec has a *yes* answer. This means that $|S| = k + 1$ because otherwise, the induced subgraph would contain more than $h$ color $\sigma$ edges as $t$ is connected to all other vertices. Furthermore, in order to achieve the density of $\alpha \geq \frac{\binom{k}{2}+k}{k+1}$, there have to be $\binom{k}{2}$ edges between the nodes $S \setminus \{t\}$ and $k$ edges from $t$ to the nodes $S \setminus \{t\}$. Because the graph $G'$ is simple, the by $S$ induced subgraph is a $(k+1)$-clique. And due to the construction, $S \setminus \{t\}$ is a $k$-clique in $G$. □

Similarly, the decision version of the at-most $h$ colored edges DSP (mHcEdgesDSPdec) is **NP**-complete. We now show the hardness for the at-least $h$ colored edges variant.

THEOREM 9. *The decision version of the at least* **h** *colored edges version (*aLHcEdgesDSPdec*) is* **NP**-*complete.*

PROOF. We show the result for the special case $\pi = 2$. The decision version of aLHcEdgesDSPdec asks to decide if there is a subset $S$ such that the induced subgraph has at least $h$ color $\sigma$ edges and a density of at least $\alpha$. It is clearly in **NP**. We use again a reduction from $k$-clique.

Given an instance $(G, k)$ of $k$-clique, we construct the following instance of aLHcEdgesDSPdec:

- Let $n = |V(G)|$ and let $K'$ be the complete graph $K_{n^4}$ in which all edges are colored 1. Furthermore, we construct $G'$ as in the proof of Theorem 8. The graph $H$ is the union of $K'$ and $G'$.

- Moreover, we set $\sigma = 1$, $h = \binom{n^4}{2} + k$, and $\alpha = \frac{\binom{n^4}{2}+\binom{k}{2}+k}{n^4+k+1}$.

Let $S \subseteq V(H)$ be a witness of aLHcEdgesDSPdec. (i) First note that $K'$ is completely in $G(S)$. Assume that $K'$ is not completely in $G(S)$ and let $T$ be the nodes from $K'$ that are in $S$, then $T$ provides $\binom{|T|}{2}$ color 1 edges. Thus we need to choose $h - \binom{|T|}{2}$ color 1 edges from $G'$, which is impossible because $h - \binom{T}{2} > n$ when $|T| < n^4$.

(ii) Next, we show that $|S \cap V(G')| = k+1$. Because $|S| \geq n^4+k+1$ and $n^4$ nodes belong to $K'$, the number of nodes in $S' = S \cap V(G')$ is at least $k+1$. Now, we show that $|S'| \leq k+1$, by showing that adding additional edges beside the necessary $k$ edges would decrease the density $d(S)$. Let $\beta = d(S')$, $s = |S'|$, $e_k = |E(K')|$, and $v_k = |V(K')|$. We can express the density $d(S)$ as follows

$$d(S) = \frac{\beta \cdot s + e_k}{s + v_k}.$$

If we add another node $u \in V(G)$ to $S'$, we also add new edges into the induced subgraph $G(S')$. Let $p$ be the number of additional edges. Then the density changes to

$$d(S \cup \{u\}) = \frac{\beta \cdot s + e_k + p}{s + v_k + 1}.$$

Note that by adding vertex $u$, we also add a new edge of color 1 between $u$ and $t$. We show that $d(S) - d(S \cup \{u\}) > 0$, hence adding additional nodes and, therefore, additional edges of color 1 leads to decreased density.

$$d(S) - d(S \cup \{u\}) = \frac{\beta \cdot s + e_k}{s + v_k} - \frac{\beta \cdot s + e_k + p}{s + v_k + 1} > 0 \quad (1)$$

$$\Leftrightarrow \beta s^2 + \beta s v_k + \beta s + e_k s + e_k v_k + e_k \quad (2)$$

$$- (\beta s^2 + \beta s v_k + e_k s + e_k v_k + p s + p v_k) > 0 \quad (3)$$

$$\Leftrightarrow \beta s + e_k - p(s + v_k) > e_k - p(s + v_k) > 0 \quad (4)$$

To see why the last inequality holds we upper bound $p$ with $n^2$ and $s$ with $n$ such that we have

$$\binom{n^4}{2} - n^2(n + n^4) > 0$$

which holds for $n \geq 2$.

Now let $G$ contain a $k$-clique $C$. Hence, by setting $S = C \cup \{s\}$, we have at least $h$ edges with color $\sigma = 1$, and we obtain a total density of $\alpha$.

If aLHcEdgesDSPdec has a *yes* answer, then it holds that $\alpha = \frac{\binom{n^4}{2}+\binom{k}{2}+k}{n^4+k+1}$ due to the previous observations (i) and (ii). Specifically, $|S| = n^4 + k + 1$ because otherwise, the induced subgraph would contain more than $h$ color $\sigma$ edges, which would reduce the density. In order to achieve the density of $\alpha$, there need to be $\binom{k}{2} + k$ edges in $G'$, which means that the nodes in $G'$ are completely connected. Hence, there is a $k$-clique in $G$. □

PROOF OF THEOREM 2. Given a graph $G = (V, E)$, $h \in \mathbb{N}$, and $\alpha \in \mathbb{R}$, the decision version asks if there there exists a densest subgraph with at least $h$ edges and density at least $\alpha$. It is clearly in **NP**. We use a reduction from the decision version of the at least $k$ nodes DSP, which asks for a densest subgraph with at least $k \in \mathbb{N}$ nodes and density of at least $\beta \in \mathbb{R}$. For our reduction, we construct a family of at most $|V|^2$ instances of the at least $h_i$ edges densest subgraph problem with $h_i = i$ for $i \in [|V|^2]$ and $\alpha = \beta$. We show that the at least $k$ nodes instance has a *yes* answer iff. one of the at least $h_i$ edges instances has at least $k$ nodes and a density of at least $\beta$.

First we show that if the at least $k$ nodes instance has a *yes* answer, one of the at least $h_i$ edges instances has at least $k$ nodes and a density of at least $\beta$. Let $\phi$ be the minimum number of edges over all graphs that are densest subgraphs of $G$ with at least $k$ nodes. Because $1 \leq \phi \leq |V|^2$ there is a a least $\phi$ edges densest subgraph instance. Let $S$ be an optimal solution for the at least $\phi$ edges DSP and assume it would not be optimal for the at least $k$ nodes DSP. Let $S'$ be optimal for the latter with the density of $d^* \geq \beta$. Note that $|E(S')| \geq \phi$ which leads to a contradiction to the optimality of $S$. Hence, $d(S) \geq d^* \geq \beta$.

For the other direction, it is clear that if there is a solution $S$ to one of the at least $h_i$ edges instances that has at least $k$ nodes and a density of at least $\beta$ then $S$ is also a solution to the at least $k$ nodes instance. □

## A.2 Proofs of Section 4

PROOF OF LEMMA 2. Assume $S$ is optimal for the at least $k$-nodes DSP but not optimal for the at least $h$-edges DSP. Let $S'$ be optimal for the latter. Now because $|S'| \geq k$ and $d(S') > d(S)$, we have a contradiction to the optimality of $S$. □

Before we prove Theorem 7, we introduce two lemmas.

**Lemma 3.** *Algorithm 1 and Algorithm 2 can be adapted to everywhere sparse multigraph input $M$.*

Proof. To adapt Algorithm 1 and Algorithm 2 for use with the multigraph $M$, we need to make the following adjustments: In Algorithm 1 and Algorithm 2, we use the general lower bound of nodes $\ell(h, \pi)$ instead of $\ell(h)$. Furthermore, in Algorithm 2, when adding an edge to $M(S_i)$, ensure that all parallel edges between a pair of nodes are added. It is noteworthy that Lemma 2 remains applicable to the multigraph $M$. This follows directly from the subgraph definition and density. Consequently, we can conclude that the adapted Algorithm 1 is optimal for the multigraph $M$.

To demonstrate that Algorithm 2 provides a $O(1)$-approximation for the multigraph $M$ we need to establish the following:

(1) If $M(S_i)$ does not contain at least $h$ edges, the addition of parallel edges introduces fewer new nodes compared to the addition of simple edges. Therefore, it can only enhance the density of the resultant subgraph.

(2) Regardless of edge colors, a multigraph can be treated as a weighted graph where edge weights correspond to the number of parallel edges between a pair of nodes $(u, v)$. As a result, the 3-approximation algorithm introduced by [2] is applicable to multigraphs. This implies that the assumption $c_1|E(S)| \geq |E(S^*)| \geq h$ used in the proof of Theorem 4 is valid.

The combination of points (1) and (2) concludes the proof. □

**Lemma 4.** *Algorithm 3 gives an $O(1)$-approximation for ATLEASTH-EDGESDSP on everywhere sparse multigraph input $M$.*

Proof. To adapt Algorithm 3 for use with a multigraph input, we should modify the peeling procedure in line 6 as follows: when we remove edges during the peeling procedure, we should remove all parallel edges that are incident to node $v_i$.

With arguments analogous to those presented in points (1) and (2) of the proof for Lemma 3, we can conclude that the theorem remains valid when applied to multigraphs. Furthermore, the running time complexity of Algorithm 3 remains unchanged. □

Proof of Theorem 7. Together with Lemma 4, the proof can be adapted from the proof of Algorithm 4, following the same line of reasoning that adding all parallel edges can only enhance the density of the resulting subgraph. The running time increases to $O(n+m\cdot\pi)$ due to the transformation of the graph to the multigraph. Note that $\pi$ is often a small constant. □

## B ILP FORMULATIONS

For the exact solution of ATLEASTHEDGESDSP, we solve the following ILP where guess the number of nodes $\ell(h) \leq k \leq n$ and return the solution with the maximum density. We use $y_u \in \{0, 1\}$ to denote if vertex $u \in V$ belongs to the solution $S \subseteq V$. Moreover, we use $x_{uv} \in \{0, 1\}$ for each edge $e = \{u, v\} \in E$ to encode if $e$ is in the densest subgraph ($x_{uv} = 1$) or not ($x_{uv} = 0$). We can relax $x_{uv} \in \{0, 1\}$ to $x_{uv} \in [0, 1]$ because we maximize over $x$ and due to Equation (7).

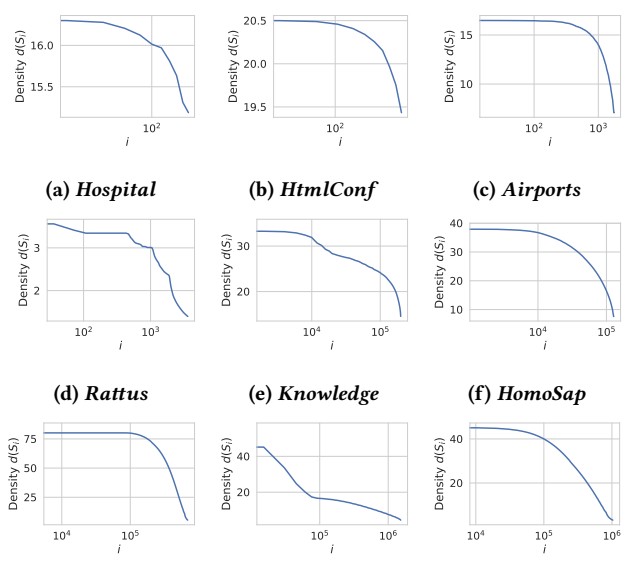

(a) *Hospital*  (b) *HtmlConf*  (c) *Airports*

(d) *Rattus*  (e) *Knowledge*  (f) *HomoSap*

(g) *Epinions*  (h) *DBLP*  (i) *Twitter*

**Figure 7: The density computed with ATLEASTHAPPROX for increasing numbers of required edges.**

$$\max \sum_{\{u,v\} \in E} \frac{x_{uv}}{k} \tag{5}$$

$$\text{s.t.} \sum_{\{u,v\} \in E} x_{uv} \geq h \text{ and } \sum_{u \in V} = k \tag{6}$$

$$x_{uv} \leq y_u \text{ and } x_{uv} \leq y_v \quad \text{for all } \{u, v\} \in E \tag{7}$$

$$x_{uv} \in [0, 1] \quad \text{for all } \{u, v\} \in E \tag{8}$$

$$y_u \in \{0, 1\} \quad \text{for all } u \in V \tag{9}$$

We can solve the at least $h$ colored edges DSP (ALHCEDGESDSP) by adding the following additional constraint:

$$\sum_{\{u,v\} \in E_c} x_{uv} \geq h_c \quad \text{for all } c \in [\pi], \tag{10}$$

where $E_c \subseteq E$ is the subset of edges with color $c \in [\pi]$.

## C DATA SET DETAILS

- *AUCS* contains relations among the faculty and staff within the Computer Science department at Aarhus University [32]. The five layers correspond to the following relationship types: current working relationship, repeated leisure activities, regular lunch, co-authorship of publication, and facebook friendship.
- *Hospital* and *HtmlConf* are face-to-face contact networks between hospital patients and health care workers [41], and between visitors of a conference [24]. The networks spans five and three days, respectively, represented by the colors.
- *Airports* is an aviation transport network containing flight connections between European airports [8]. The layers represent different airline companies.
- *Rattus* contains different types of genetic interactions of *Rattus Norvegicus* [13]. The six layers are physical association,

direct interaction, colocalization, association, additive genetic interaction defined by inequality, and suppressive genetic interaction defined by inequality.

- *Knowledge* is based on the *FB15K-237 Knowledge Base* data set [39]. The FB15K-237 data set contains knowledge base relation triples and textual mentions of the Freebase knowledge graph entity pairs. We represent the entities as nodes and different relations colored edges.
- *HomoSap* is a network representing different types of genetic interactions between genes in Homo Sapiens [18].
- *Epinions* is an online social network and general consumer review site. Members on the platform have the option to determine whether or not they trust one another [30].
- *DBLP* is a subgraph of the DBLP graph [31] containing only publications from $A$ and $A^*$ ranked conferences (according to the Core ranking). The edges describe collaborations between authors, and the edge colors represent the different conferences.
- *Twitter* is a subgraph of the Twitter graph representing users and retweets in the period of six months before the 2016 US Presidential Elections [37]. Each edge represents a retweet and is labeled as factual or misinformation.
- *FriendFeed* and *FfTwYt* are based on the former *FriendFeed* social media aggregation service that allowed users to consolidate and view updates from various social networking platforms and websites [14]. The *FriendFeed* dataset contains interactions among users collected over two months. The three layers represent commenting, liking and following. *FfTwYt* is a network in which users registered a Twitter and a Youtube account associated to their Friendfeed account.

Received 20 February 2007; revised 12 March 2009; accepted 5 June 2009

