# OpenReview forum: "Finding Densest Subgraphs with Edge-Color Constraints"
_ACM.org/TheWebConf/2024/Conference — TheWebConf24 Oral_

### Official Review · Reviewer_nw9E · 2023-11-03

**Novelty:** 5
**Technical Quality:** 6

**Review:**

This paper introduces a new problem of finding the densest subgraph under specific constraints on graphs with labeled edges (labels are associated with colors). In more details, it defines the problem of finding the densest subgraph that has at least h nodes of a given color, and also has introduces other variants: find the densest subgraph having at most, and exactly h edges of the given color. The paper studies the complexity of these problems, showing that they are NP-complete. For the problem of finding a densest subgraph with at least h colored edges, the paper provides a linear-time constant factor approximation  algorithm (0(1) approximation) when the input graph is sparse. In the empirical evaluation, the approximation quality is studied but only in small graphs (since we cannot have the best solution in large graphs). The characteristics and running times are also studied for different algorithms that have been provided by the paper. Experiments also argue the necessity of the proposed methods by showing that the basic DSP would identify subgraphs with distribution of edge colors which is different from the overall distribution.
Strong points:
- The paper introduces a new problem with a clear definition
- It provides a thorough theoretical study of the complexity of the problem and designs several algorithmic solutions for different cases
- The designed algorithms provide constant approximation factor
- Experiments show detailed quantitative study

However, there are some weak points:

- Applicability of the problem in real world applications: It is not clear whether the proposed formulation is more relevant than some similar existing problems and algorithms when addressing real world problems. The paper needs to make it clearer why one would use this method instead of handling its data analysis with other methods such as (among many other possibilities): Boden, Brigitte, et al. "Mining coherent subgraphs in multi-layer graphs with edge labels." Proceedings of the 18th ACM SIGKDD international conference on Knowledge discovery and data mining. 2012.
--> While I understand the technical difference between formulations, it is not clear whether this new formulation is more appropriate for some settings. In fact, the examples that have been provided in the introduction are not really convincing.
- While the papers provides a guarantee of constant approximation factor for its algorithms, it is not clear how good is this approximation factor in large graphs. As we cannot study this empirically in large graphs (impossible to identify the best solution), the only possible solution may be to study theoretically this point.
- Qualitative results are missing: While the paper provides a detailed quantitative empirical evaluation, it does not provide concrete examples of results. Particularly, it would have been interesting to show interesting subgraphs that have been identified by this method but not by existing ones, and explain why these specific subgraphs are interesting.
typos:
- 273 the by V' induced subgraph G(V')

**Questions:**

- Would it be possible to provide more concrete examples where the introduced problem is relevant (more relevant than existing methods, for example, compared to multilayer graph formulation)?
- It is really nice that the designed algorithms provide constant approximation factor. Would it be possible to identify the approximation factor (or an upper bound on it)?
- Can you provide examples of interesting subgraphs that have been identified by your approach and not by other possible "similar" alternatives of formulating the problem (existing formulations)?

**Ethics Review Description:**

No issue

**Reviewer Confidence:**

3: The reviewer is confident but not certain that the evaluation is correct

**Scope:**

4: The work is relevant to the Web and to the track, and is of broad interest to the community

---

### Official Review · Reviewer_jt9v · 2023-11-15

**Novelty:** 6
**Technical Quality:** 7

**Review:**

This paper studies the problems of finding dense subgraphs that contain exactly/ at most/ at least h_i edges of color h_i in graphs with single or multiple edge colors.
The paper proves that the decision versions of these three problems are NP-complete even for graphs with two colors, by reduction from the k-clique problem.
Then, it presents a linear-time constant-factor approximation algorithm for everywhere sparse graphs with a single color per edge.
The algorithm is based on an \alpha-approximation algorithm for the at least k-node DSP problem.
Finally, it shows how the proposed algorithms can be adapted to solve the problems in the case of multiple colors per edge.
The algorithms are evaluated in 12 real-world networks and compared against two baselines.
Results prove that:
- more than 93% of the instances of ATLEASTHEDGESDSP are optimally solved or solved within 1% error;
- the running time of the approximation algorithm for ATLEASTHEDGESDSP is at most 19 seconds for the largest dataset;
- more than 70% of the instances of ALHCEDGESDSP are optimally solved or solved within 1% error;
- the running time of the approximation algorithm for ALHCEDGESDSP is at most 21 seconds for the largest dataset;
- the approximation algorithm for ALHCEDGESDSP can find denser subgraphs with at least one edge per color than the competitor MLDS.

The only limitation I encountered in this paper is the fact that the "exactly h_i" and the "at most h_i" problems are only briefly covered.

**Questions:**

1. e in Definition 1 is not defined. I think the paper meant "at least h_i edges e with i \in c(e)".

2. According to the definition of \mathbb{N}, Definition 1 requires the output subgraph to contain at least one edge for each color. Do the algorithms proposed extend to the case where some h_i = 0?

3. The definition of \sigma in the proof of Theorem 8 is missing. In addition, it is not clear why h is not a vector even though the proof assumes that \pi = 2.

4. The paper could discuss if running [34] in the dual graph of the input graph could solve ALHCEDGESDSP. Here, I assume that the dual graph G* of G=(V,E) has node set equal to E, and an edge between each pair of e_1,e_2 \in E that share an endpoint in G.

5. Section 4.3 defines M multiple times.

6. "Figure 2 shows the results for AUCS and FriendFeed" -> Figure 2 shows also the results for FfTwYt.

7. TYPO: "in Figure 3a and Figure 3b show the color distributions" -> "Figure 3a and Figure 3b show the color distributions".

**Ethics Review Description:**

.

**Reviewer Confidence:**

3: The reviewer is confident but not certain that the evaluation is correct

**Scope:**

4: The work is relevant to the Web and to the track, and is of broad interest to the community

---

### Official Review · Reviewer_bsVr · 2023-11-21

**Novelty:** 6
**Technical Quality:** 6

**Review:**

The paper considers finding densest subgraphs in graphs with categorical edge attributes. Given a condition on edge colors (attributes), the paper considers the at least, exactly, at most variants; and also studies a problem where an input vector on constraints is provided.

+ The paper provides a new approach for diverse dense subgraph finding based on categorical edge attributes. In particular, the assumption about the everywhere sparse graphs seem to work well for theoretical bounds.

- Presentation needs to be improved. One simple thing that can help a lot is changing the alg abbreviations and making it consistent throughout the paper. Another thing is increasing the fonts of tables and figures.

- In evaluation, all the graphs are said to be "everywhere sparse". One natural question is what happens on a graph that is not everywhere sparse. Another question is how exactly one can test if the input graph is everywhere sparse.

- Sec 4.3 is too terse, to the point of being unclear. It's not clear how the application of proposed algorithms can naturally handle the multi-edge graphs. The technique proposed in this part is also not experimentally evaluated.

**Questions:**

See above.

**Reviewer Confidence:**

4: The reviewer is certain that the evaluation is correct and very familiar with the relevant literature

**Scope:**

4: The work is relevant to the Web and to the track, and is of broad interest to the community

---

### Official Review · Reviewer_7q56 · 2023-11-21

**Novelty:** 6
**Technical Quality:** 6

**Review:**

In this paper, the authors study the problem of diverse densest subgraph detection. They show the NP-Hardness of the corresponding problems and propose efficient approximation algorithms with theoretical guarantees. They evaluate the performance of the proposed algorithm and the results demonstrate the efficiency and effectiveness of the proposed algorithms.

Strength:
S1. New approximate algorithms with theoretical guarantees are proposed.
S2. Experiments are conducted and the results demonstrate the effectiveness of the proposed algorithms.


Weakness:
W1. The real applications of the proposed algorithms is unclear.
W2. The impact regarding assumption on everywhere sparse graphs is unclear.
W3. The related work on multiple graphs should be discussed more.

**Questions:**

Q1. The real applications of the proposed algorithms is unclear. Although several potential application scenarios are mentioned in Section 1, it is better if real applications with citation can be provided. Moreover, it is more convincing if some case studies can be provided.

Q2. It seems that the everywhere sparse graphs assumptions is too strong. In the power-law graphs, some parts of the graphs are very dense.

Q3. Following the experiments, most of the datasets are multilayer graphs. It is better if more related work on multilayer graphs can be discussed.

**Reviewer Confidence:**

4: The reviewer is certain that the evaluation is correct and very familiar with the relevant literature

**Scope:**

4: The work is relevant to the Web and to the track, and is of broad interest to the community

---

### Official Review · Reviewer_dL1Z · 2023-11-24

**Novelty:** 4
**Technical Quality:** 5

**Review:**

The paper proposes and studies a new problem: given a graph with each edge having one color, and a vector h of color requirements, the problem aims to find the densest subgraph with exactly, at most, and at least h[] colored edges for each color in h. The paper analyzes the hardness of this problem and proposes a linear time algorithm with an approximation guarantee of density. Experimental results demonstrate the effectiveness of the algorithm. Nevertheless, the motivation can be enhanced and the technique novelty is around the bar.

Strong points:

S1. The overall presentation is clear and logical, including the problem definition, the techniques, etc.

S2. The hardness of the problem is proved rigorously.

S3. The related works are well-surveyed and discussed.

S4. The design of the experiments is reasonable and the datasets are from multiple sources.

Opportunities for improvement:

O1. Although the densest subgraph problem is extensively studied, its advantage over other dense subgraph models is not quite clear. For instance, applications on large dynamic data may prefer the k-core model which is more efficient. An in-depth discussion/comparison is needed.

O2. The technique novelty is shadowed by existing algorithms. While the algorithm proposed in this paper effectively addresses the presented problem, it relies on computing DSP based on the k-core model [2]. We may discuss/evaluate other dense subgraph models, e.g., k-truss.

O3. On finding the densest subgraph with “exactly/at most” h[] colored edges, there should be more descriptions and discussions on extending the proposed algorithm for “at least” h[] colored edges.

**Questions:**

Please refer to O1-O3.

**Reviewer Confidence:**

3: The reviewer is confident but not certain that the evaluation is correct

**Scope:**

3: The work is somewhat relevant to the Web and to the track, and is of narrow interest to a sub-community

---

### Decision · Program_Chairs · 2024-01-22

**Decision:**

Accept (Oral)

**Comment:**

The reviewers generally liked this paper. The new problem formulation as a variant of densest subgraph was interesting. The theoretical analysis of the proposed algorithm was clean and well-presented. The experiments were detailed and convincing. There were some concerns about the real-world applicability of the proposed problem formulation, but all the reviewers felt that the positives outweighed the negatives. I recommend acceptance.